# One-Dimensional Computational Model of Gyttja Clay for Settlement Prediction

Grzegorz Kacprzak, Artur Zbiciak, Kazimierz Józefiak , Paweł Nowak and Mateusz Frydrych *

Faculty of Civil Engineering, Warsaw University of Technology, 16 Armii Ludowej Ave., 00-637 Warsaw, Poland
* Correspondence: mateusz.frydrych.dokt@pw.edu.pl; Tel.: +48-22-2346543

**Abstract:** One of the most important subjects of geomechanics research is finding mathematical relationships which could correctly describe behavior of the soil under loading. Safety of every engineering structure depends strongly on accuracy and correctness of this description. As laboratory tests show, macroscopic properties of soil are complicated. Therefore, working out appropriate load-settlement relationships is considered to be a very difficult geomechanics tasks to solve. A majority of constitutive models proposed to date concern mineral soils and there is very little research related to modelling organic soil behavior under loading. In case of organic soils, due to their very complicated and composite structure, constitutive models are often formulated empirically based on laboratory tests of particular soils. The authors of this paper propose a 1-D rheological structure which accounts for complex behavior of soil related to the settlement process. The model simulates immediate reversible elastic settlement and plastic soil deformation as well as primary and secondary (creep effect) consolidation. Material parameters of the model were determined by a curve fitting procedure applied for a natural scale settlement test of plate foundation. The test was carried out in soil conditions connected with Eemian geological structure of Warsaw, i.e., Eemian glacial tunnel valley in Warsaw called Żoliborz Glacial Tunnel Valley filled with organic soils being up to 20 metres thick. This area has lately become an object of interest of investors as a site for building construction.

**Keywords:** soil rheology; settlement; creep; plate foundation; geotechnical engineering

## 1. Introduction

One of the most important topics of geomechanics research is to find equations which will correctly describe the behavior of the loaded soil. Accuracy and correctness of description of loaded soil work, determines the safety of each structure. Due to a very complex and non-homogeneous structure of soil, the elaboration of accurate relationships load—settlement is considered to be one of the most difficult geomechanics task, especially in case of organic soils.

The necessity of erecting and maintaining building structures on an organic subsoil requires proper prediction of ground surface deformations with the use of geotechnical parameters. In the premises of Warsaw, the geological and engineering problems of organic soils are mostly related to organic carbonate deposits filling the glacial tunnel valley paleodoline the so-called Żoliborz Glacial Tunnel Valley, which extends meridionally Warsaw from the Żoliborz to the Okęcie District. Settlement of structures located in Warsaw within the glacial tunnel valley result primarily from settlement of gyttja, including its consolidation, which occur during the construction and operation phases of buildings. Gyttja, as a calcareous and organic mud, is formally classified as low-bearing capacity soil. This definition should rather refer to young Holocene organic lake sediments characterized by low geotechnical parameters with the texture of 'slime' or 'ooze' [1] as described by Hampus von Post in 1862. Geological history of 100 ka (kilo annum) changed geotechnical properties by consolidation and other different post-depositional processes sensu lato compacting scrutinized soil (i.e., moisture variation not caused by consolidation, cementation, time

factor, etc.). A soil consolidation process is complicated and generally involves three stages: (i) initial compression, (ii) primary consolidation (pore pressure dissipation), (iii) secondary consolidation (creep rheological effect) [2]. All the three phases do not need to proceed one after another. Organic soils contain all three elements of total settlement in the initial phase of loading. A secondary consolidation phenomenon is still not fully understood.

The process of consolidation of highly compressible soils involves such phenomena as immediate deformation of the bubbles of gaseous water in the pores and deformation of the skeleton under the influence of effective stress [3–7]. This type of deformation is assumed to be elastic. The second one is related to the decrease volume of soil and gradual soil consolidation. The duration of consolidation under constant loading depends on soil permeability. In addition, there are secondary deformations, which are the result of long-term structural deformations of the soil (creep). The rate of these deformations depends on the rheological properties of the soil (viscosity); the greater the structural viscosity of the soil, the longer the skeleton creep process persists.

The results of research carried out by many researchers [3,6–9] indicate that the characteristics describing the process of organic soil consolidation is non-linear, which makes it difficult to use them in computational methods. The current state of research surrounds the topic presented in this article for several reasons. These are matters related to the very problem of consolidation based on non-classical rheological systems [10]. This non-linearity results not only from the change in the state of the soil, but also from the large anisotropy of stress in the subsoil and its variability in the deformation process [9]. In addition, the high compressibility of organic soils makes it necessary to take into account variable ground geometry during consolidation calculations, which leads to non-linear geometrical relationships in numerical solutions. The deformations which are more complicated than mineral soils, require the use of calculation methods based on complex soil models taking into account the different behavior of organic soils under load. Of great interest is the use of numerical analysis based on theoretical rheological models, the use of which provides adaptability to other geotechnical systems. A good example is the very fresh work on Non-Darcian flow and rheological consolidation of saturated clay [11]. Therefore, it is necessary to correctly estimate parameters or deformation characteristics of soils describing particular stages of the deformation process. The characteristics defining the deformation process influence the value of the parameter which depends on the stresses and time for a given type of soil. The values of deformation properties of the subsoil adopted for further analyses determine the stiffness of the subsoil under the direct foundation, as well as the stiffness of piles, displacement of columns or barettes included in the deep foundations.

Organic soils vary from the mineral soils primarily by a significant content of the organic substance (exceeding 2%) and, in most of cases, coloideality of the liquid phase. In this case study geological model covered gyttja with organic matter ranging mostly from 10% to 30%, with other components like calcium carbonate $CaCO_3$ (between 10–50% of the content) and other mineral (non-carbonate) and non-organic parts (in the range 28–68% of the content). Due to a very complex and non-homogeneous structure, in case of organic soils, the empirical models developed during the laboratory research of the soil samples formulation is the most common. The authors of this thesis decided to create a rheological model of the organic soil based on the results of a large scale test (scale 1:1), which is the sample loading of the foundation plate settled directly on the ground. The test was performed in soil conditions connected with Eemian geological structure of Warsaw, i.e., Eemian glacial tunnel valley in Warsaw called Żoliborz Glacial Tunnel Valley filled with organic soils with the thickness up to over 20 m. The theory of a time rate of one-dimensional consolidation was first proposed by Terzaghi [11]. It was based on the assumption that a relationship between effective stress and strain is linear elastic and it describes only the primary consolidation process. Thereafter, several investigators [12–17] used visco-elastic models to study one-dimensional consolidation, i. e. infinite soil layer of known thickness loaded at the top. For example Gibson & Lo [15] proposed a rheological structure for a soil skeleton which consisted of Hookean spring connected in series with

a Kelvin-Voigt element where effective stress was calculated using Terzaghi's continuity equation. [17] used a Kelvin-Voigt element in effective stress space but with a non-linear dashpot. It is also possible to model soil settlement using more complicated rheological structures in total stress space where a certain dashpot accounts for Terzaghi's consolidation theory [2].

In this paper the authors propose a one-dimensional rheological model of plate foundations on gyttja clay during consolidation, defined in total stress space, taking into account all three phases of the process. The current state of knowledge describes the curves resulting from theoretical considerations, including using statistical analysis, for many reasons [18]. The choice of the serial connection of particular element was specified, as recalled, by the infiltration of consolidation phases. The model was not previously used by engineers for settlement prediction. The main feature of the model consists of an original set of non-linear explicit ODEs defining the evolution of the rheological structure. The set of constitutive relationships is strongly non-linear and should be integrated with the use of classical algorithms for solution of ordinary differential equation systems with respect to time.

## 2. Load Test in Natural Scale

In order to define model parameters of the investigated soil, a sample loading test was carried out. The test was performed on a concrete raft with the dimensions 5.0 m × 5.0 m × 0.30 m at depth of 3.0 m below the ground level, after the removal of ground layers on the non-reinforced subsoil. The geotechnical cross-section pertaining to the issue under consideration, along with the soil and water conditions, is shown in Figure 1. CPT soundings were particularly useful in correlating the results. Figure 1 shows the results of only the cone resistance for the CPT sounding and the pressure results for the DMT sounding, i.e., the most relevant results from the point of view of the conducted research. The authors decided to present only the most relevant data for the sake of legibility of the presented results.

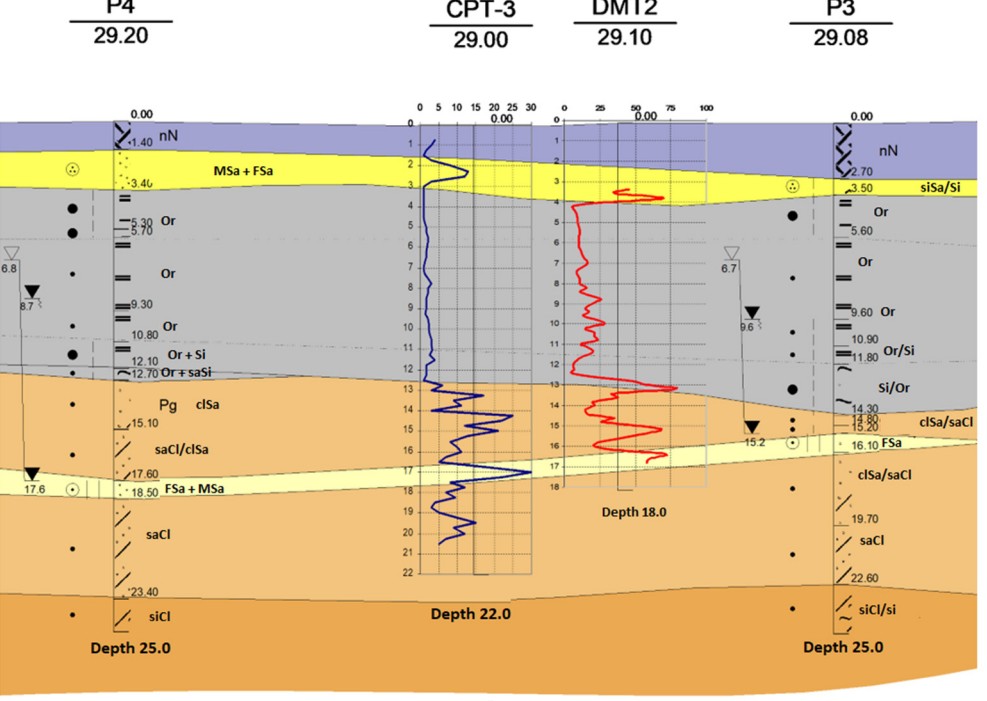

**Figure 1.** Geotechnical cross-section pertaining to the issue at hand.

The results of the CPT investigation are shown in Table 1. The documentation compiled was quite extensive, but only the most relevant information from the point of view of the

article was selected, so the first culm of Table 1 shows the results for many soundings. $S_u$ stands for shear strength in a undrained shear test.

**Table 1.** CPT probe results for the issue at hand.

| CPT Number | Depth[m] | LL | Su[kPa] |
|---|---|---|---|
| CPT–S1 | 2.4–8.8 | 0.2 | 45 |
| CPT–S2 | 3.0–13.0 | 0.2 | 50 |
| CPT–S3 | 2.8–12.8 | 0.2 | 60 |
| CPT–S4 | 3.0–13.0 | 0.25 | 55 |
| CPT–S5 | 3.4–11.2 | 0.25 | 50 |
| CPT–S6 | 3.0–9.6 | 0.22 | 50 |
| CPT–S5b | 11.2–13.8 | −0.05 | 200 |
| CPT–S6 | 3.0–9.6 | 0.22 | 50 |
| CPT–S7 | 4.6–9.2 | 0.22 | 60 |
| CPT–S8 | 3.2–14.2 | 0.15 | 70 |
| CPT–S9 | 3.4–15.0 | 0.15 | 70 |
| CPT–S10 | 3.3–9.0 | 0.18 | 60 |

The observations were conducted during a 1–1.5 month period. The sample loading was placed stage by stage, with the up-to-date measurements. A 300 t ballast was placed and, as a result, stress of 120 kPa was obtained. To control the settlement of concrete slab, 5 supervising benchmarks were stabilized in each corner (benchmark 1 to 4) and in the middle of the raft (benchmark 5). Positioning of the benchmarks is presented in Figures 2 and 3. The analysis carried out was very complicated, and measurements from the middle repertory (benchmark no. 5) were selected for simulation in order to keep the height of the results transparent.

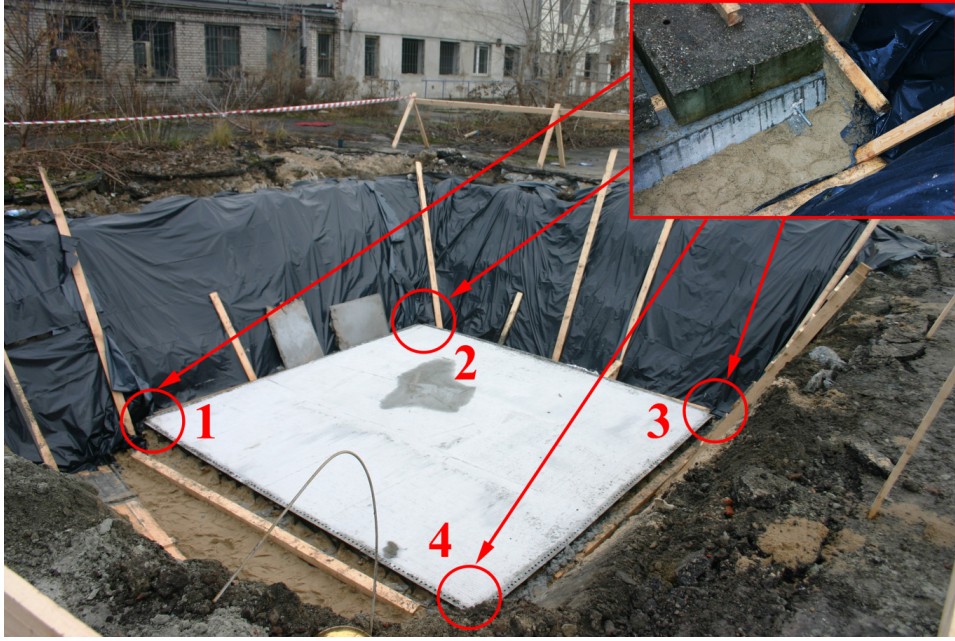

**Figure 2.** Positioning of benchmarks no. 1–4.

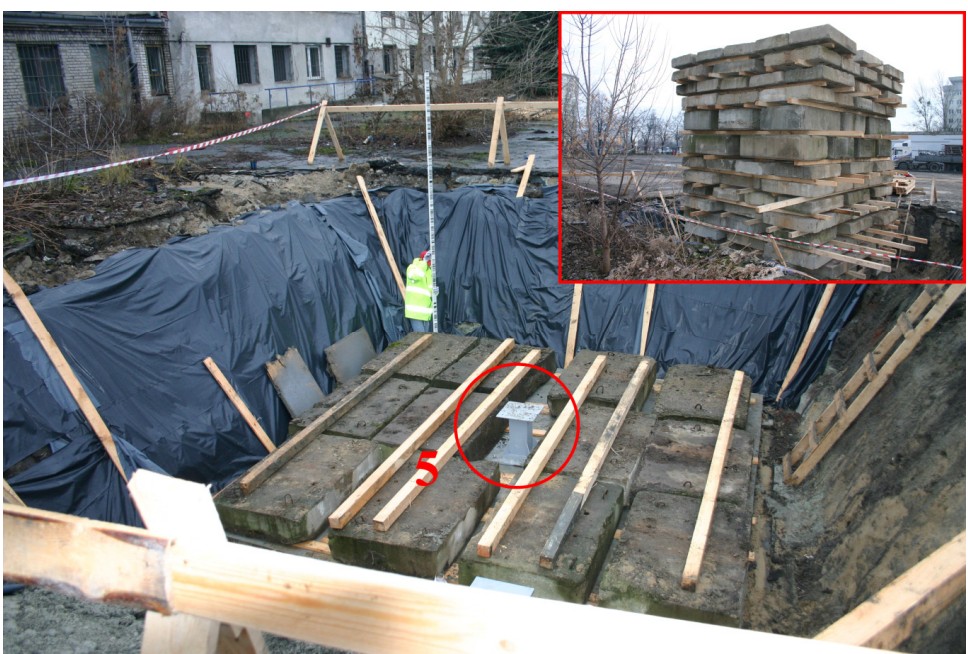

**Figure 3.** Positioning of benchmark no. 5 and concrete plates view.

The measuring network consisted of 5 supervising benchmarks and 3 reference benchmarks. The measurement was executed using the precise levelling method, with the automatic leveler LEICA NA 3003 and the set of invar levelling rods. The initial measurement of the settlement of the concrete raft was taken before the plate was loaded. The following measurements were done successfully after loading the raft with a layer consisted of several concrete plates (Figure 3, right corner), and then during the removal of the plates. At the time of the measurement, the temperature was between 0 °C and 8 °C.

Calculations of the vertical settlement values were performed using a strict method submitting both the primary (initial) and secondary (up-to-date from the following days) measurements into equalization, conducting the analysis of the constancy of the reference benchmarks. The average error of evaluation of the benchmarks' settlement was given to indicate with what precision (accuracy) the slab settlement measurements were taken $m_{dH} \leq 0.5 \, mm$.

## 3. Rheological Model of Settlement

### 3.1. Constitutive Differential Equations in 1D

The rheological structure, shown in Figure 4, was used to obtain constitutive differential equations. As in a settlement analysis vertical displacement of a foundation should be found, the authors assumed stress-displacement relation in the rheological structure instead of stress-strain relation. This resulted in obtaining the constitutive equations between vertical displacement and vertical total stress. In the model, there are four main elements connected in series. The displacements of every section are indicated in Figure 4.

The material parameters $C\alpha$ ($\alpha$ = 1, 2, 3) are elastic proportionality constants. Their units are $N/m^3$ since a stress-displacement relation was chosen. Constants $C\alpha$ are introduced assuming that there is no constraints concerning lateral strains, i.e., boundary conditions are similar to Young's modulus definition. The material constants $D_{\beta}$ ($\beta$ = 1, 2) are soil viscosities and their units are $N \cdot s/m^3 = Pa \cdot s/m$. All the material parameters are defined in the total stress space.

The topmost spring element (displacement $s^e$) simulates the elastic behaviour according to the equation

$$s^e = \frac{\sigma}{C_1} \tag{1}$$

This element accounts for immediate reversible change in displacement after applying the load.

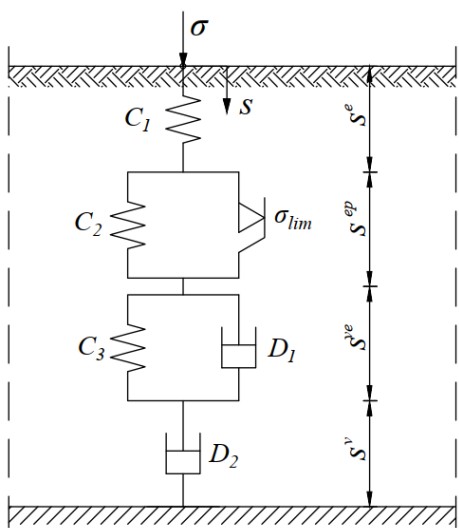

**Figure 4.** Soil settlement rheological structure.

The next section is an elastic-plastic element in which the displacement $s^{ep}$ occurs only if the stress exceeds stress limit $\sigma_{lim}$ [N/m$^2$]. This element accounts for plastic soil deformation. The following equation governs the elastic-plastic displacement rate $\dot{s}^{ep}$:

$$\dot{s}^{ep} = \begin{cases} \frac{C_1}{(C_1+C_2)\sigma^p} \left[ \sigma^p \left( \dot{s} + \frac{C_3}{D_1} s^{ve} - \frac{\sigma}{D_{eq}} \right) \right]^+ & if \ |\sigma^p| \geq \sigma_{lim} \\ 0 & if \ |\sigma^p| < \sigma_{lim} \end{cases} \tag{2}$$

where

$$\sigma^p = \sigma - C_2 s^{ep} \tag{3}$$

and $[\cdot]^+$ denotes a projection onto the set of non-negative numbers

$$[z]^+ = \begin{cases} z & if \ z > 0 \\ 0 & if \ z \leq 0 \end{cases} \quad \forall z \in R \tag{4}$$

It should be emphasized that the model is strongly non-linear taking into account both viscoelastic and plastic properties of the soil material. The paper presents an original set of constitutive equations formulated in the explicit form via the non-linear ODE. Such equations cannot be found in the literature. Formulation of Equation (2) needs the notion of associated flow rule along with Kuhn-Tucker conditions [19]. In case of such approach a crucial point is to evaluate the so called Lagrange multiplier which defines the rate of the elastic-plastic displacement (denoted $\dot{s}^{ep}$ in the paper). Having the value of the Lagrange multiplier, the set of constitutive equations has a simple analytical form as presented in the paper.

Moving back to the analysis of the rheological scheme shown in Figure 4 let us note that the two bottom sections of the model contain dashpot elements which are related to the consolidation process. The bottom dashpot is responsible for irreversible creep displacement which is connected with secondary consolidation. The visco-elastic displacement rate $\dot{s}^{ve}$ and viscous displacement rate $\dot{s}^{v}$ can be calculated as in the case of Newtonian fluids

$$\dot{s}^{ve} = \frac{1}{D_1} (\sigma - C_3 s^{ve}) \tag{5}$$

$$\dot{s}^{v} = \frac{\sigma}{D_2} \tag{6}$$

The four displacement components (Figure 4) are connected in series and individual displacements should be added up to obtain the total displacement

$$s = s^e + s^{ep} + s^{ve} + s^v \tag{7}$$

Formula (2) for the case where $|\sigma^p| = \sigma_{lim}$ was obtained taking into account that during plastic yielding

$$C_1 s^e = \sigma_{lim} + s^{ep} C_2 \tag{8}$$

which implies

$$C_1 \dot{s}^e = \dot{s}^{ep} C_2 \tag{9}$$

and by substituting Equation (7), differentiated with respect to time. Variable $\sigma^p$ was put into the square bracket to take into account loading and unloading cases using only one formula. Equations (1), (2), (5)–(7) form a differential-algebraic equation system which can be numerically solved with given material constants ($C_\alpha$, $D_\beta$, $\sigma_{lim}$), loading function $\sigma(t)$ and initial conditions to obtain settlement values in time. The material parameters were determined in this paper based on the test loading experiment results. The initial conditions are

$$s(0) = 0; \ s^e(0) = 0; \ s^{ep}(0) = 0; \ s^{ve}(0) = 0 \tag{10}$$

An analytical solution of the system exists only for a simple instantaneous loading shown in Figure 5

$$\sigma(t) = \begin{cases} \frac{F_{max}}{A} & if \ t \geq 0 \ and \ t < t_0 \\ 0 & otherwise \end{cases} \tag{11}$$

in the form [17]

$$s = \sigma_0 \varphi \ \text{where} \ \varphi = \begin{cases} \varphi_1 & if \ t < t_0 \\ \varphi_2 & if \ t > t_0 \end{cases} \tag{12}$$

where settlement functions $\varphi \ \varphi_1$ and $\varphi \ \varphi_2$ depend on material parameters and are defined as follows:

$$\varphi_1 = \frac{1}{C_1} + \frac{A_1}{C_2} + \frac{t}{D_2} + \frac{1}{C_3}\left[1 - exp\left(-\frac{t}{\lambda}\right)\right] \tag{13}$$

$$\varphi_1 = \frac{A_2}{C_2} + \frac{t_0}{D_2} + \frac{1}{C_3}exp\left(-\frac{t}{\lambda}\right)\left[exp\left(\frac{t_0}{\lambda}\right) - 1\right] \tag{14}$$

where $\lambda = \frac{D_1}{C_3}$ and

$$A_1 = \left[1 - \frac{F_{gr}}{F}\right]^+ \tag{15}$$

$$A_2 = min\left(\frac{F_{gr}}{F}, \left[\left[1 - \frac{F_{gr}}{F}\right]^+\right]\right) \tag{16}$$

and symbol $[\cdot]^+$ is defined in Equation (4).

For the loading shown in Figure 5, assuming A = 25 m$^2$ and F$_{max}$ = 2512 kN and material parameters determined by optimization for the test loading experiment (Table 2), the model was solved numerically and using analytical formulae. There was no difference between a numerical and analytical solution. The results of this simple creep test are shown in Figure 6.

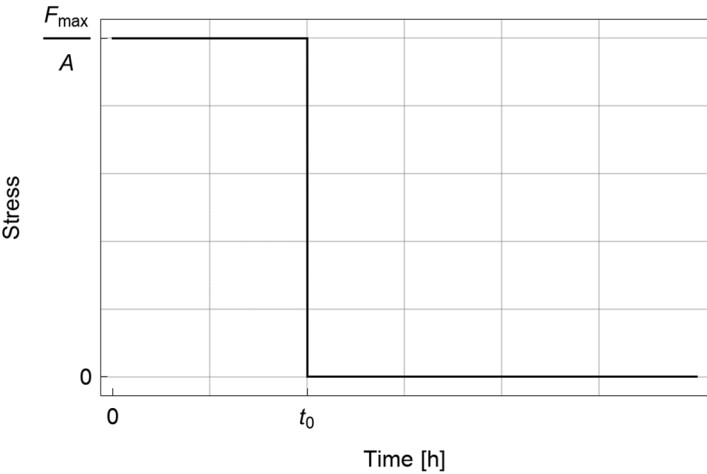

**Figure 5.** Instantaneous loading for creep test simulation.

**Table 2.** Material parameters of the settlement model determined based on the test loading experiment by curve fitting.

| $C_1 \left[\frac{kPa}{m}\right]$ | $C_2 \left[\frac{kPa}{m}\right]$ | $C_3 \left[\frac{kPa}{m}\right]$ | $D_1 \left[\frac{kPa}{m}\right]$ | $D_2 \left[\frac{kPa}{m}\right]$ | $\sigma_{lim} [kPa]$ |
|---|---|---|---|---|---|
| 5495 | 25,145 | 5765 | $4.12 \cdot 10^6$ | $4 \cdot 10^7$ | 40 |

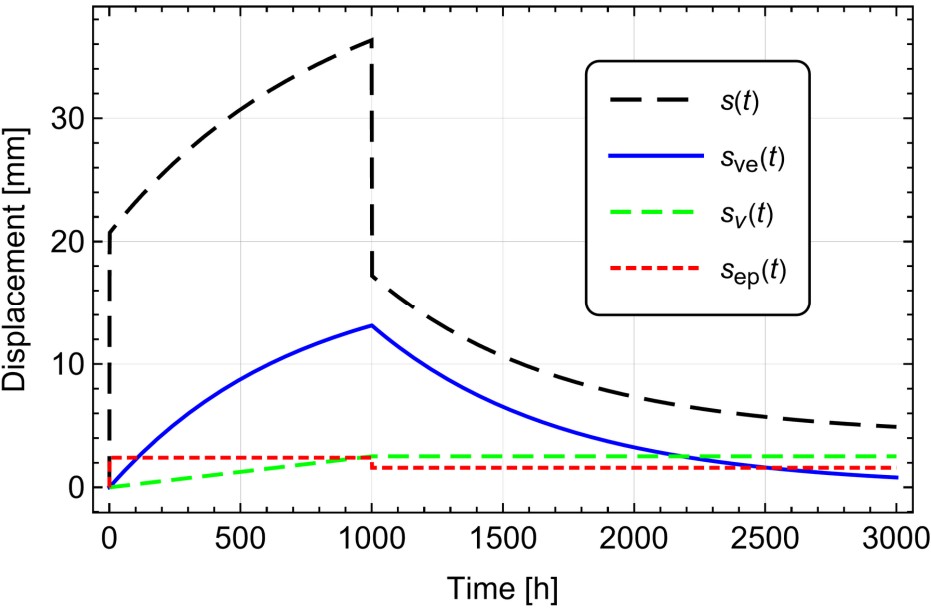

**Figure 6.** Creep test simulation results.

Material parameters of the model can be found by curve fitting to a plate foundation test loading experiment. Such a field test is carried out for a foundation of a certain shape and dimensions. In this paper, it has been done for a square plate with 5.0 m long sides. Material parameters of the model were determined by curve fitting procedure applied for settlement test results. To estimate model parameters for different foundation dimensions, the authors propose to use an elastic solution for deflection due to a uniformly loaded flexible area. It is a consolidation model in which there is only one spring element in effective stress space and the relationship for settlement can be written as [20,21]

$$s = \frac{q\omega B \left(1 - v^2\right)}{E} \tag{17}$$

where $B$ denotes foundation width or diameter, $\omega$ is a coefficient that depends on a foundation shape and stiffness, $E$ is effective soil deformation modulus related to normal consolidation or swelling line, and $\nu$ is Poisson's ratio. For deflection under the foundation center, $\omega$ ranges from 1.0 for a circular foundation to 4.0 for a rectangle plate for which $L/B = 100$. For a square foundation $\omega_{sq} 1.12$. Having a set of parameters determined by curve fitting $\{C_\alpha^{cf}, D_\beta^{cf}, \sigma_{lim}^{cf}\}$ from a test loading experiment for a foundation characterized by $\omega^{cf}$ and $B^{cf}$, the model parameters of any foundation shape can be calculated as follows (see Equation (17)):

$$C_\alpha = B^{cf} \omega^{cf} \frac{C_\alpha^{cf}}{\omega B}, \ \alpha = 1, 2, 3 \tag{18}$$

$$D_\beta = B^{cf} \omega^{cf} \frac{D_\beta^{cf}}{\omega B}, \ \beta = 1, 2 \tag{19}$$

$$\sigma_{lim} = B^{cf} \omega^{cf} \frac{\sigma_{lim}^{cf}}{\omega B} \tag{20}$$

For the experiment used for optimization in this paper $\omega^{cf} = \omega_{sq} = 1.12$ and $B^{cf} = 5.0 \ m$.

### 3.2. Influence of the Width of Foundation on the Value of Settlements

In case of calculation of large plates, the definition of the depth of an active settlement zone becomes a problem.

According to the literature [21–25], the value of the settlement increases proportionally to the increase of the size of the foundation. While calculating the foundations of large dimensions (big footings, large foundation plates) the active zone theoretically becomes very large, which leads to difficult and time-consuming calculations. The solution of this problem may be a method of division of the plate into smaller sections and considering its settlement separately. In such case the influence on the settlement of the adjoining foundation should be taken into account. According to [26] there is a boundary dimension of the equivalent spread footing. When it is exceeded, the settlement of the foundation does not increase, and the equation for the calculation of the settlement of the foundation can be presented as:

$$s = \frac{\sigma_0^2}{E_0 \gamma} \sqrt{\frac{3}{2\pi}} \left( \frac{\gamma B}{\sigma_0} \right) \left[ 2 - \sqrt[3]{\frac{3}{2\pi\beta}} \sqrt[3]{\frac{\gamma B}{\sigma_0}} \right] \tag{21}$$

Relationship (21) is the function of independent variable $\frac{\gamma B}{\sigma_0}$, the course of which is presented in Figure 7.

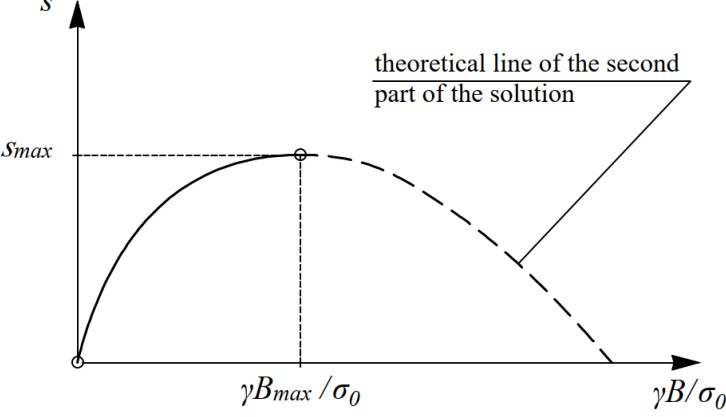

**Figure 7.** Graph of the settlement function $= s\left( \frac{\gamma B}{\sigma_0} \right)$.

From the graph, it can be read out that the settlement of the foundation increases with the growth of the dimension of foundation B. It can be indicated that function (21) reaches its maximum in point $\gamma B_{max}/\sigma_0$, which shows that in order to calculate the settlement of a large plate, it is enough to calculate the settlement for a given load, for a square slab with side dimension $\sigma_0/\gamma$. Therefore, the research was conducted in the natural scale in-situ on the square plate with the dimensions approximate to the equivalent square footing, resulting from the established range of loading $\sigma_0$ = 120 kPa and the soil weight $\gamma$ = 21 kN/m³. The dimension of the equivalent square footing obtained in order to calculate accurately the settlement of the large plate is $B_{max} = 120/21 = 5.7\ m$.

### 3.3. Influence of the Width of Foundation on the Value of Settlements

Material constants of the proposed settlement model were determined by optimization using data of the plate foundation test loading described in previous sections. Figure 8 shows the stress applied to the plate foundation in the course of the field test. Linear interpolation was assumed between test points. Having an interpolated function it was possible to numerically solve a differential-algebraic system of equations for the given values of material constants.

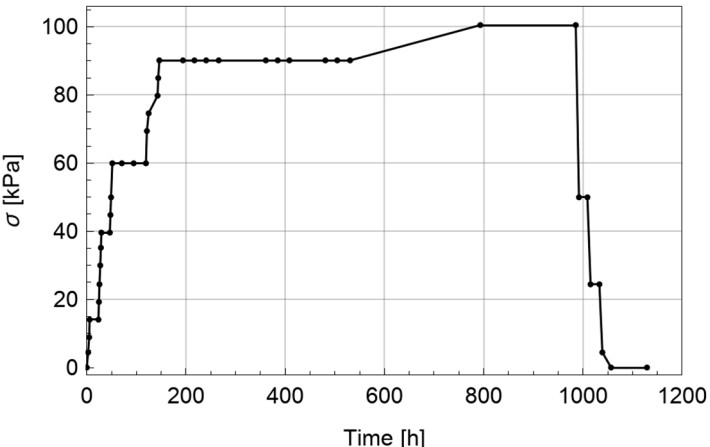

**Figure 8.** Stress under the plate foundation during the settlement field test, linearly interpolated between measurement points.

The optimal set of material parameters for the rheological model indicates such values of parameters $\tilde{p}$ for which functional $F(p)$ reaches its minimum. Thus, the minimization problem can be stated as follows:

$$\tilde{p} = arg\min_{p \in \Omega} F(p) \tag{22}$$

where $\Omega$ denotes the set of admissible parameters' values.

The optimization problem defined by Equation (22) was solved using a least-squares method implemented in Mathematica software [27]. In this approach, the functional $F(p)$ was defined as follows:

$$F(p) := \Sigma_{j=1}^{N} \left| s_j^R(t) - s_j(t;p) \right|^2 \tag{23}$$

where $s_j(t;p)$ are the settlement values determined by the rheological model and $s_j^R$ are test data and N is the number of data points. In our case N = 41 and $p = \{C_\alpha,\ D_\beta,\ \sigma_{lim}\}$.

Nelder-Mead nonlinear optimization technique was used [28]. However, it was not possible to find the global optimum of the functional with this method and only a local optimum could be found. Although in the case of this optimization there were many local minima, Nelder-Mead algorithm was used since it was computationally efficient. Many attempts with different initial constraints had to be made to obtain correct curve fitting results. In such a non-linear problem, where only a numerical solution of a differential-

algebraic system was available, it was difficult to find a true global optimum. However, it was possible, in this case, to find a sufficient minimum (after rejecting incorrect local minima).

A determined set of parameters is shown in Table 2. Figures 9 and 10 show the optimization results and separate model displacement components, respectively. It can be observed that the model can simulate the consolidation process quite well, although some effort had to be made to find material parameters. The secondary effects are important and they take in a case of organic soil a very long time to occur. Therefore, to predict the long term settlement the proposed mathematical model needs to be validated for a wider period.

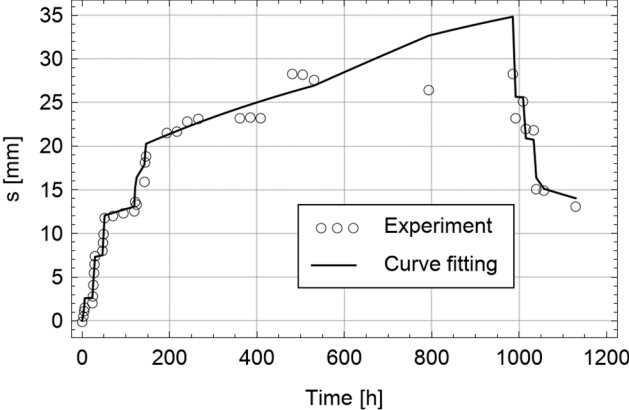

**Figure 9.** Curve fitting results. Total settlement curve.

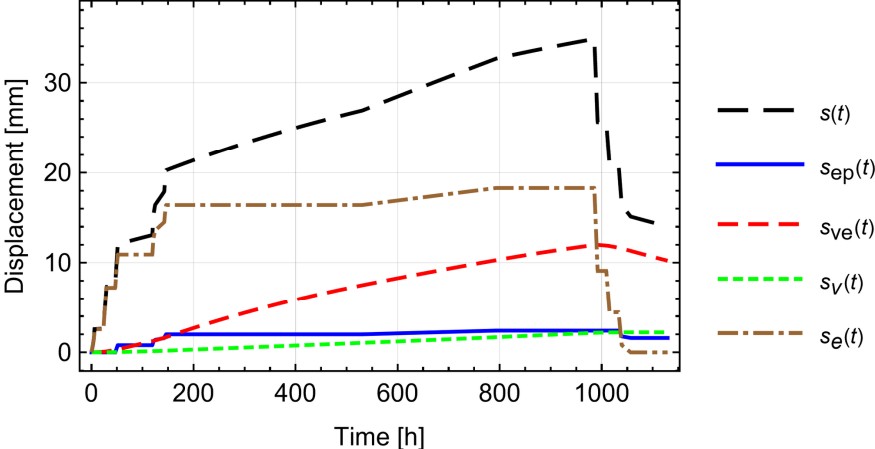

**Figure 10.** Separate displacement components of the rheological model.

## 4. Conclusions

The interpretation of the obtained values of $C_1$, $C_2$, $C_3$ defined as a stiffness modulus of the elastic subsoil by direct analogy to the Winkler—Zimmerman model (kN/m$^3$) must be carried out $k = 1/(1/C_1 + 1/C_2 + 1/C_3) = 2530$ kPa/m. This value is convergent to 4000 kPa/m as a result of field observation (120 kPa/0.03 m).

$D_1$ and $D_2$ constants define modified viscosity of Newton's liquid. In comparison with original viscosity unit expressed in kPa·s, $D_1$ and $D_2$ can be called the modified viscosity by analogy between a constant of Winkler-Zimmerman model (modulus of subgrade reaction $k$ [kPa/m]) and the modulus of elasticity E expressed in kPa (by analogy to the modulus expressed in kPa/m in the discussed case the viscosity unit has been changed into kPa s/m). Observation of the results of fitting analysis of the $D_1$ and $D_2$ values is a prompt to think that there is a logic reference to a coefficient of soil consolidation $C_v$ expressed in m$^2$/s. Assuming that the typical value of $C_v$ equals $10^{-7}$ m$^2$/s [29] in the presented case

the inverse value of $C_v$ is compatible with $D_1$ and $D_2$ values: $D_1 = 4.12 \cdot 10^6$ kPa·h/m and $D_2 = 4.0 \cdot 10^7$ kPa h/m ($1/D = 1/D_1 + 1/D_2$ gives $3.6 \cdot 10^6$), respectively.

In the case of $\sigma_{lim}$ treated as the shear strength in the undrained conditions ($s_u$ or $c_u$ according to the European Standard EN 1997), the values calculated in fitting analysis can be compared with the results of the triaxial tests $S_u = 1/2 \cdot (\sigma_1 - \sigma_3)$, which is the half values of the stress deviator q.

Figure 11 shows the results of the consolidated undrained triaxial test performed on the gyttja recovered from the depth of 6 m from the site of the real scale load test. Assuming that the submerged unit weight of gyttja is $\gamma' = 8$ kN/m$^3$ and the at-rest earth pressure coefficient $K_0$ of gyttja is 0.9, finally effective radial stress on the depth of 6 m is $\sigma_3' = 6$ m · 8 kN/m$^3$ · 0.9 = 43.2 kPa.

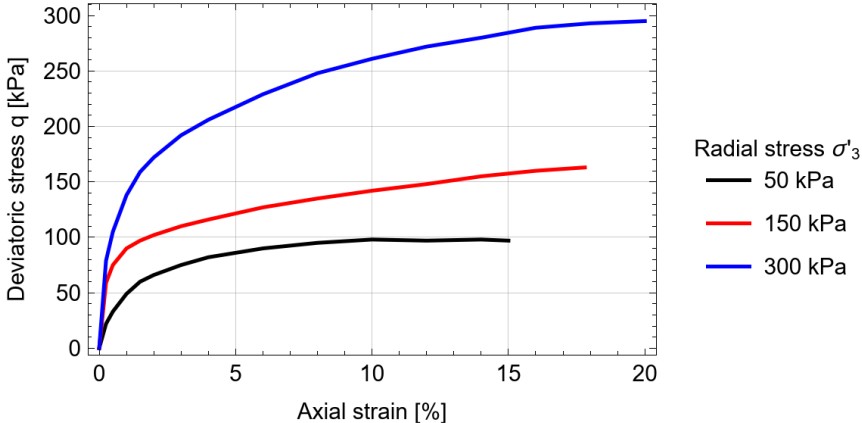

**Figure 11.** Triaxial compression test results for the gytias directly under the base plate.

The analysis of the results of triaxial test indicates satisfactory/good convergence of the shear strength $s_u$ value from the laboratory tests with the value of limit stress $\sigma_{lim}$ estimated in the curve fitting procedure. Taking into account half of the limit value of the deviator $q$ for the radial stress $\sigma_3' = 50$ kPa (nearest to 43.2 kPa) the value of shear strength is $s_u = q/2 = 98/2 = 49$ kPa which is close to $\sigma_{lim} = 40$ kPa obtained in simulation.

It should be emphasized that the methodology of formulating constitutive relationships with the use of rheological schemes applied in this paper has led to their explicit form as a system of nonlinear differential equations. A similar approach was presented by the authors in [30–32], concerning not only soils, but also asphalt-aggregate mixtures. Such approach involves applying classical and non-classical rheological elements (springs, dashpots, sliders, etc.). The same elements can be used for the description of different phenomena occurring in soils and asphalt-aggregate mixtures. For example in soils primary and secondary consolidation can be modelled. On the other hand, in asphalt-aggregate mixtures rheological structures can be applied for modelling creep and relaxation phenomena as well as binders' properties called zero shear viscosity [33].

Proper parametrization of the rheological ground model is necessary also in order to analyze dynamical behavior of buildings considering soil-structure interaction. Presented approach to determining shear strength of soil can be useful for correct assessment of the control procedure of building—soil-foundation system with plastic behavior of soil [34–36].

The curve fitting of the model (see Table 2 and Figure 9) consists of iteratively solving the system of non-linear differential equations. For that purpose, Mathematica software was applied with the program's build-in ODEs solvers. These solvers are well suited for classical continuous problems. Therefore, special attention had to be put in order to solve a discontinuous differential equation as Equation (2). During calculations the authors performed a simplified sensitivity analysis which demonstrated that the small perturbations of rheological scheme parameters do not change the character of the solution significantly. In general, it was observed that the major changes (e.g., one order of magnitude) of plasticity (slider) and viscosity (dashpot) elements result in erroneous prediction of permanent

settlement after unloading. An in-depth sensitivity analysis needs a lot more calculations. The work on this field is currently carried out by the authors.

**Author Contributions:** Conceptualization, G.K. and A.Z.; methodology, A.Z.; software, K.J.; validation, G.K. and K.J.; formal analysis, A.Z.; investigation, K.J.; resources, P.N.; data curation, K.J.; writing—original draft preparation, K.J. and M.F; writing—review and editing, G.K., A.Z., K.J., P.N. and M.F.; visualization, K.J., P.N. and M.F.; supervision, A.Z.; project administration, G.K., M.F. and P.N. All authors have read and agreed to the published version of the manuscript.

**Funding:** This paper was co-financed under the research grant of the Warsaw University of Technology supporting the scientific activity in the discipline of Civil Engineering and Transport.

**Institutional Review Board Statement:** Not applicable.

**Informed Consent Statement:** Not applicable.

**Data Availability Statement:** All data used in the article are given in the bibliography or own sources have been used in the case of photos.

**Acknowledgments:** The authors acknowledge the Warbud company for the opportunity to use the results of the field research.

**Conflicts of Interest:** The authors declare no conflict of interest.

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
