# Peer review of "One-Dimensional Computational Model of Gyttja Clay for Settlement Prediction"

_sustainability, doi:10.3390/su15031759_

Round 1

Reviewer 1 Report

Investigating the behavior of soils during loading is of great importance in guiding engineering practice. The authors structured the corresponding one-dimensional model and determined the parameters in it. The study is of some significance. However, there are still some structural and content problems with the current paper, so I think it should be revised first.

1.      The introduction is too dated and lacks some recent literature research, which makes the study less significant.

2.      Line 127. Skip directly to Figure 8 here, please make sure it is appropriate.

3.      The author should carefully check the superscript and subscript of the unit to ensure that they are correct. In addition, the font format should be uniform.

4.      Lines 171-179, Different font sizes. There are similar situations in other places.

5.      The scale of the individual displacement components in Figure 5 seems to be incorrect, please check further to make sure.

6.      With regard to the conclusion section, the author should distinguish the conclusion section from the results section. Therefore I think the author should reorganize the third and fourth parts.

Author Response

Dear Reviewer,
On behalf of the authors, I sincerely thank you for the submitted review! Thank you for your factual and substantive comments, which will help our article gain quality.
I would like to kindly inform you that all the comments you made have been corrected and can be seen in the posted text. We intentionally with the co-authors post the article with open tracking of the changes so that you can verify the changes made. First of all, the introduction has been updated, adding current research on the topic under discussion. In addition, reference was made to 2, 3 and 4 by removing redundant sentences or adding new ones keeping in mind factual and linguistic correctness. We also confirm comment No. 5 regarding the scale of displacement in Figure 5, which has also been corrected. We have also addressed the last comment by increasing the clarity of the text, which organizes the entire article, according to the authors. 

We are open to other comments, which we will address immediately.
With best regards on behalf of the authors,
Mateusz Frydrych

Reviewer 2 Report

The paper proposes a 1-D rheological structure, which accounts for the complex behavior of soil related to the settlement process, including immediate reversible elastic settlements, plastic soil deformation, and primary and secondary (creep effect) consolidation. The algorithm and model are based on field tests and existing research conclusions, and their effectiveness has also been effectively verified. The manuscript can be accepted after minor revisions.

1.       The research status elaboration part in the introduction section does not involve the latest research results and works of literature, and it is suggested to supplement it.

2.       Figures 5 and 9 are unclear, and redefining the curves' representation is recommended.

3.       In addition to comparing the results of the total settlement, is it possible to compare the components in Figure 9? For example, by means of numerical simulation.

4.       It is suggested that the part of the conclusion section involving assumptions and discussions (including Figure 10) be included in the main text.

Author Response

Dear Reviewer,
On behalf of the authors, I sincerely thank you for the submitted review! Thank you for your factual and substantive comments, which will help our article gain quality.
I would like to kindly inform you that all the comments you made have been corrected and can be seen in the posted text. We intentionally with the co-authors post the article with open tracking of the changes so that you can verify the changes made. First of all, the introduction has been updated, adding current research on the topic under discussion. In addition, all comments have been addressed by removing unnecessary sentences or adding new ones keeping in mind factual and linguistic correctness. Figures 5 and 9 were corrected increasing clarity, in addition, more data were added to Figure 9. We also addressed the last comment by increasing the clarity of the text, which organizes the entire article according to the authors. 

We are open to other comments, which we will address immediately.

All the best in New Year!

With Best Regards on behalf of the authors,
Mateusz Frydrych

Reviewer 3 Report

1. Please provide the large-scale test results for five different points.

2. Please include the soil profile and engineering soil properties for the field test.

3. It is unclear how the authors have determined the model parameters from the field test results. If the authors have used the field test results to determine the model parameters, model verification by using the field test results will be questionable. 

4. Description of a figure in the text must be before the figure. 

5. Font size of the text in lines 171 to 173 is different.

6. L174-Eq. 2: The first conditional statement seems wrong, and the equal sign must be replaced with ">=". 

7. Leave a blank space between a number and its unit.

8. L216: use "and" instead of the math symbol or put a description for the symbol.

9. Be consistent in formatting and using "Figure" and "Equation" in the text.

10. Use superscripts for powers (e.g., L206, L233, and L341).

11. L351: Change "cv" to "Cv".

12. Fix the problems related to punctuation and grammar.

Author Response

Dear Reviewer,
On behalf of the authors, I sincerely thank you for the submitted review! Thank you for your factual and substantive comments, which will help our article gain quality.
I would like to kindly inform you that all the comments you made have been corrected and can be seen in the posted text. We intentionally with the co-authors post the article with open tracking of the changes so that you can verify the changes made. 
First and foremost, the deficiencies regarding the soil parameters and scale studies have been corrected, adding better quality drawings. Removed all problems related to comments 4-12 by removing unnecessary sentences, letters or adding new ones keeping in mind the factual and linguistic correctness. Increased the correctness of the English language along with checking mathematical formats. 

We are open to other comments, which we will address immediately.

All the best in New Year!

With Best Regards on behalf of the authors,
Mateusz Frydrych

Round 2

Reviewer 1 Report

As author has made changes, I think it can be accepted

Author Response

Dear Reviewer,
On behalf of the authors, we sincerely thank you for accepting the changes we have made. Once again, thank you for providing substantive and very good comments, which allowed us to increase the quality of our article.

We sincerely thank you.

Kind regards,

Mateusz Frydrych with co-authors

Reviewer 3 Report

1. Figure 1: The acronyms used in this figure are unclear (what do they stand for?).

2. Table 1: What is the meaning of the first column? It is not clear.

3. Table 1: Use decimal points instead of commas.

4. Table 1: Does Su stand for undrained shear strength? It is not clear.

5. Figure 1 and Table 1: It is expected to see tip-resistant and skin friction graphs for a typical CPT test. It is unclear where they are and what the purple and red plots in Figure 1 are.

6. Please include the measured settlements of benchmarks 1 through 5 versus time.

7. L165: What does mdH stand for?

8. L223: p in the variable must be superscript.

9. L247: The figure does not have any caption.

10. L373 & L374: "v" in "Cv" must be subscript.

11. L278: Leave a blank space between "1.12" and "and"; also between the numbers and units.

11. L384: Change "m3" to "m3".

Author Response

Dear Reviewer,
On behalf of the authors, we sincerely thank you for providing factual and substantive comments to enhance the quality of our article. We have addressed all the comments by adding the appropriate provisions in the article or increasing the readability of the posted data. Explanatory sentences have been added regarding Figure 1 and Table 1 (su). In addition, comma designations were changed to decimal designations. An explanatory sentence was also added for the CPT and DMT studies included in the article. The authors intentionally included only a snippet of the required studies in order to maintain the highest possible quality of the presented content from the very extensive documentation for the entire project. Thanks to the reviewer's attention, the article has gained clarity and clearly presents the content. In the case of Note 6, a sentence was added to explain the measurements used - the results of the middle repertory, which best represents the research conducted, were used (we have included a graph showing the results of all measurements). Clarification was also added for mdH, which deals with the precision of surveying measurements. The comments in lines L223 and L373, L374, L278 and L384 were also corrected. Thankyou for this remarks.

All changes were carried out by the authors in "tracking" mode so that it is easy to verify the changes made. Note 9 concerns the purpose of the introduced equations, which are i.e. integration constants resulting from arithmetic, which will be given as definitions.

Attention was also paid to the English language and corrections were made to give the article quality.

Kind regards,

Mateusz Frydrych
